# Role of Calcimimetics in Treating Bone and Mineral Disorders Related to Chronic Kidney Disease

**DOI:** 10.3390/ph15080952

**Published:** 2022-07-31

**Authors:** Yi-Chou Hou, Cai-Mei Zheng, Hui-Wen Chiu, Wen-Chih Liu, Kuo-Cheng Lu, Chien-Lin Lu

**Affiliations:** 1Division of Nephrology, Department of Medicine, Cardinal-Tien Hospital, School of Medicine, Fu-Jen Catholic University, New Taipei City 24205, Taiwan; athletics910@gmail.com; 2Division of Nephrology, Department of Internal Medicine, Shuang Ho Hospital, School of Medicine, College of Medicine, Taipei Medical University, New Taipei City 11031, Taiwan; 11044@s.tmu.edu.tw; 3TMU Research Centre of Urology and Kidney, Taipei Medical University, New Taipei City 11031, Taiwan; leu3@tmu.edu.tw; 4Graduate Institute of Clinical Medicine, College of Medicine, Taipei Medical University, New Taipei City 11031, Taiwan; 5Department of Medical Research, Shuang Ho Hospital, Taipei Medical University, New Taipei City 11031, Taiwan; 6Department of Biology and Anatomy, National Defense Medical Center, Taipei 11490, Taiwan; wayneliu55@gmail.com; 7Section of Nephrology, Department of Medicine, Antai Medical Care Corporation, Anti Tian-Sheng Memorial Hospital, Pingtung 92842, Taiwan; 8Division of Nephrology, Department of Medicine, Taipei Tzu Chi Hospital, Buddhist Tzu Chi Medical Foundation, School of Medicine, Buddhist Tzu Chi University, Hualien 97004, Taiwan; 9Division of Nephrology, Department of Medicine, Fu-Jen Catholic University Hospital, School of Medicine, Fu-Jen Catholic University, New Taipei City 24205, Taiwan; janlin0123@gmail.com

**Keywords:** calcimimetics, calcium-sensing receptors, CKD–MBD, fibroblast growth factor 23, secondary hyperparathyroidism

## Abstract

Renal osteodystrophy is common in patients with chronic kidney disease and end-stage renal disease and leads to the risks of fracture and extraosseous vascular calcification. Secondary hyperparathyroidism (SHPT) is characterized by a compensatory increase in parathyroid hormone (PTH) secretion in response to decreased renal phosphate excretion, resulting in potentiating bone resorption and decreased bone quantity and quality. Calcium-sensing receptors (CaSRs) are group C G-proteins and negatively regulate the parathyroid glands through (1) increasing CaSR insertion within the plasma membrane, (2) increasing 1,25-dihydroxy vitamin D3 within the kidney and parathyroid glands, (3) inhibiting fibroblast growth factor 23 (FGF23) in osteocytes, and (4) attenuating intestinal calcium absorption through Transient Receptor Potential Vanilloid subfamily member 6 (TRPV6). Calcimimetics (CaMs) decrease PTH concentrations without elevating the serum calcium levels or extraosseous calcification through direct interaction with cell membrane CaSRs. CaMs reduce osteoclast activity by reducing stress-induced oxidative autophagy and improving Wnt-10b release, which promotes the growth of osteoblasts and subsequent mineralization. CaMs also directly promote osteoblast proliferation and survival. Consequently, bone quality may improve due to decreased bone resorption and improved bone formation. CaMs modulate cardiovascular fibrosis, calcification, and renal fibrosis through different mechanisms. Therefore, CaMs assist in treating SHPT. This narrative review focuses on the role of CaMs in renal osteodystrophy, including their mechanisms and clinical efficacy.

## 1. Introduction

Chronic kidney disease (CKD) is characterized by a chronic decrease in glomerular filtration and chronic structural damage to the kidney [1]. The decrease in glomerular filtration engenders multiple comorbidities, as well as the loss of nephrons, resulting in complications, including fluid overload, electrolyte imbalance, hypertension, and hormonal dysregulation; such dysregulation leads to complications such as insufficient production of erythropoietin, vitamin D, or phosphaturic hormones [2,3]. Such comorbidities may also result in multiple organ dysfunction syndromes. Renal osteodystrophy—which is characterized by impaired bone remodeling caused by the interplay between vitamin D deficiency, elevated parathyroid hormone (PTH) level, and uremic toxin accumulation due to a decline in glomerular filtration—is common in patients with CKD and end-stage renal disease (ESRD), and its associated vascular calcification and fracture may negatively influence the quality of life or even lead to mortality [4]. Calcimimetics (CaMs), which bind to the calcium-sensing receptors (CaSRs) of the parathyroid gland (PTG), have been used to treat primary or secondary hyperparathyroidism (SHPT). Furthermore, the pleiotropic effects induced by CaMs may engender improvements in other systems apart from the skeletal system [5]. This article presents a review of the pathogenesis of renal osteodystrophy and the pleiotropic effects of CaMs in renal osteodystrophy treatment.

## 2. Pathogenesis of CKD–Mineral and Bone Disorder: PTH–Vitamin D Interaction Dysregulation as Glomerular Filtration Declines

Phosphate is strongly regulated by phosphaturic hormones, including fibroblast growth factor 23 (FGF-23) and PTH [6,7]. Physiologically, with the assistance of a skeletal tissue buffer, the kidney excretes approximately 800 mg of phosphate ingested daily. The sodium–phosphate cotransporter within the proximal renal tubule transports the excreted phosphate from the glomerulus through an interaction with phosphaturic PTH–FGF-23/klotho complex signaling [8]. As the glomerular filtration rate (GFR) declines, a compensatory increase in phosphaturic hormone production occurs, which regulates the function of the remaining nephrons in maintaining phosphate excretion. To regulate the production of PTH, the CaSRs on the parathyroid glands modulate the synthesis and release of PTH, the concentration of which is inversely correlated with the plasma calcium concentration [9]. Moreover, the concentration of active vitamin D—that is, 1,25-dihydroxy vitamin D (1,25(OH)2D)—derived from the kidney decreases in tubule interstitial cells. Persistent parathyroid gland hyperplasia directly activates the expression of 1-α hydroxylase within the renal tubules to maintain the production of active vitamin D and downstream intestinal absorption in the intestine. However, FGF-23 reduces vitamin D production by directly inhibiting kidney 1-α hydroxylase. The resulting vitamin D deficiency reduces intestinal calcium absorption and CaSRs mediated PTH inhibition.

Physiologically, bone remodeling is essential for bone health. Osteoblasts are the main cells responsible for bone formation, because they interact with each other to form units of bone called osteons [10]. Mesenchymal and skeletal stem cells differentiate into osteoprogenitor cells, pre-osteoblasts, and then osteoblasts. The osteoblasts are embedded into the mineralized bone matrix and, finally, evolve into osteocytes [11]. In bone, osteoclasts remove mineralized bone (bone resorption) to facilitate the formation of a bone matrix by osteoblasts (i.e., bone formation) [12]. Osteocyte-lining cells recruit osteoclasts through an interaction between the receptor activator of nuclear κB (RANK) and the RANK ligand (RANKL). Multiple signals are involved in bone resorption activation, including PTH signals. PTH activates osteoblasts to release the RANKL in mature osteoblasts and suppress its decoy receptor osteoprotegerin (OPG) production in early osteoblasts, which further activates the osteoclasts [13] and induces osteoblastic differentiation and maturation through the downregulation of osteocyte-derived sclerostin and upregulation of Wnt/β–catenin signaling [14]. The interaction of the osteoblasts and osteoclasts governs bone turnover, which is reflected in the osteoid volume per bone volume and the osteoid maturation time [15,16]. In the various stages of CKD, the bone remodeling process may be disrupted by several complications. However, in most cases of renal osteodystrophy with a high bone turnover, the bone is roughly in balance because of the high bone formation accompanied by high resorption rates. For renal osteodystrophy with a low bone turnover rate, such as adynamic bone disease, low bone formation by indolent osteoblasts is accompanied by low bone resorption rates.

### 2.1. Early-Stage CKD–Mineral and Bone Disorder: Low-Bone-Turnover Disease

In patients with early CKD (stages 1–2), the serum concentration of Wnt signaling inhibitors, such as DKK1 (Dickkopf-1), SOST (sclerostin), and sFRP (secreted frizzled-related proteins), increases because of the release from osteocyte or extraosseous-calcifying tissue and the decrease in renal excretion [17]. The Wnt-signaling inhibitor acts on the osteoblast (OB), resulting in decreased OB viability. In addition, the retention of protein-bound uremic toxins such as indoxyl sulfate (IS)/p-cresol sulfate (PCS) further attenuated the OB and OC viability and function. Uremic osteoporosis was used to describe the effects of protein-bound uremic toxins on bone quality loss with normal bone mass [18].

Coen et al. conducted a cross-sectional study to compare the bone turnover rates between different stages of CKD based on GFR estimates [19]. They observed that the fraction of woven osteoid decreased as the GFR increased. Barreto et al. also demonstrated the difference in bone histomorphometry between CKD stages 2 and 3 and CKD stages 4 and 5. Specifically, they observed that decreases in the osteoid and osteoblast surface areas reflected decreased bone formation and resistance to bone remodeling [20]. Barreto et al. reported that the indoxyl sulfate concentration was the main factor contributing to low bone turnover independent of vitamin D deficiency or PTH concentration [20]. As a protein-bound uremic toxin, indoxyl sulfate can induce PTH resistance in osteoblasts by generating oxidative stress [21]. Indoxyl sulfate also inhibits osteoclast differentiation after cellular entry through organic anion transporters [22]. Furthermore, indoxyl sulfate inhibits osteoclastogenesis by regulating the nuclear factor of activated T-cell cytoplasmic 1 through the aryl hydrocarbon receptor [23]. In addition to uremic toxin accumulation, insulin resistance and deficiency could lead to a decrease in osteoblast proliferation and, thus, degrade bone formation [24,25].

### 2.2. Advanced CKD–Mineral and Bone Disorder: High-Bone-Turnover Disease Due to SHPT

The development of high-turnover bone disease occurs only later in CKD, when the serum PTH levels can overcome the peripheral resistance to PTH and other inhibitory factors in bone formation. When the renal functions progress (>stage 3) to calcium, phosphate, vitamin D, and PTH dysregulation, patients may present with high or low PTH levels. High PTH levels drive the indolent OB into high viability and function but poor quality in behavior, resulting in both bone quality and quantity loss [18]. In advanced CKD, secondary hyperparathyroidism (SHPT) accelerates bone remodeling and thereby induces osteitis fibrosa, which is characterized by the loss of mineralization and the formation of woven osteoids [26]. Osteitis fibrosa is mainly due to SHPT, and mixed uremic osteodystrophy is characterized by a lack of mineralization most often attributed to vitamin D deficiency. Hypocalcemia mediated by vitamin D deficiency can intensify the sensitivity of the parathyroid glands to vitamin D by reducing CaSR expression. When uremia occurs, the circadian rhythm of the parathyroid glands is dysregulated, and signaling pathways that govern cell proliferation, such as the mammalian target of rapamycin complex 1, are activated by ribosomal protein S6 or the TGF-α/epithelial growth factor receptor [27] and induce parathyroid nodular hyperplasia [28,29,30]. The persistent secretion of PTH is due to peripheral PTH resistance and PTG hyperplasia, which will enhance bone resorption. Increased bone destruction induces the release of calcium and phosphate from bone tissue into the extraosseous tissue. Consequently, calcium apatites may be deposited in the vascular system and promote vascular calcification.

### 2.3. Advanced CKD–Mineral and Bone Disorder: Low Bone Turnover Disease Due to Medical or Surgical Parathyroidectomy or Aluminum Intoxication

In the medical or surgical treatment of SHPT with the removal of the stimulator of PTH, the bone cells may return to their innate low bone cell viability status, the low bone turnover disorders. We are using renal osteodystrophy for CKD patients’ bone diseases [18].

The low-turnover bone diseases include osteomalacia, aluminum-induced bone disease, and adynamic bone disease (ABD). ABD becomes common in dialysis centers with the increasing use of calcium and potent vitamin D analogs to suppress PTH. PTH resistance is unique to CKD and occurs because of PTH receptor downregulation and osteoblast dysfunction [31] by persistently elevated PTH, excessively low 1,25-dihydroxy vitamin D levels [32], and accumulated protein-bound uremic toxins. In low-turnover bone disease, disturbed bone remodeling occurs with decreased bone resorption and bone formation. Osteoblast apoptosis is a major mechanism that suppresses bone formation [33] [34]. The rate of bone microdamage repair is reduced in low-turnover CKD patients [35] and causes clinical fractures.

## 3. Role of CaSRs in SHPT

CaSRs are present in different systems, including osseous tissues, cardiomyocytes, parathyroid glands, immune cells, and vascular smooth muscle cells. A CaSR is a group of class C G-protein-coupled receptors that govern calcium homeostasis [36]. PTH release is regulated through minute changes in serum calcium concentrations (1.1–1.3 mM) [37]. The expression of CaSRs is directly regulated by CaSR gene promoters through 1,25(OH)2D, glial cells missing-2 (GCM2), or various extracellular changes in calcium-mediated proinflammatory cytokines [38,39,40,41]. Vitamin D deficiency in CKD induces hypocalcemia as a result of the decreased intestinal absorption of calcium [42]. Prolonged hypocalcemia will decrease the inhibitory effects of CaSRs on PTG and thereby induces parathyroid hyperplasia [43]. Furthermore, hypocalcemia modulates PTH secretion through posttranslational modification [44]. In CKD, PTH nodular hyperplasia may be intensified by the dysregulation of the molecular circadian rhythm of Gcm2 and other proteins, such as cyclin D1 [28]. The associated downregulation of CaSRs reflects further parathyroid gland proliferation [45]. In different calciotropic tissues, such as the parathyroid glands or intestine, CaSR homodimers also regulate cell proliferation, differentiation, or apoptosis (Figure 1).

### 3.1. Action of CaMs

#### 3.1.1. The Pharmacodynamics of CaMs

CaMs are ligands that potentiate the activity of CaSRs. These ligands can be categorized as agonist (type I) or allosteric (type II) modulators [5]. The binding affinity of intracellular calcium with CaSRs is low (EC_50_: 1.25 mM); however, CaMs, with their allosteric binding, have a high capacity to rapidly activate CaSRs (EC(50) = 34 nM). Thus, CaMs can reduce PTH secretion by approximately 30-fold compared with calcitonin [46]. The available CaMs include cinacalcet and etelcalcetide [47,48], which bind to the intramembranous and extracellular domains of CaSRs, respectively [49].

CaSRs comprise three domains: a conserved extracellular Venus flytrap (VFT) domain, a cysteine-rich domain, and a heptahelical transmembrane (7TM) domain [50]. Physiologically, calcium can serve as an orthosteric ligand and reduce the secretion of PTH by interacting with the CaSR VFT domain. However, hypercalcemia due to the excessive intestinal absorption of calcium or excessive vitamin D receptor analog use is common in CKD and ESRD [51]. Persistent hypercalcemia induces extraosseous calcification and vascular calcification, which are associated with relatively high mortality in patients with CKD and ESRD [52]. Therefore, CaMs may be less likely to induce hypercalcemia when interacting with CaSRs, and they may serve as positive allosteric modulators in the 7TM domain, which potentiates sensitivity to calcium in the VFT domain [53,54].

#### 3.1.2. The Factors Influencing the Potency of CaMs 

Several factors influence the potency of action of CaMs. Hyperphosphatemia, which is common in patients with CKD and ESRD, can inhibit CaSR activity. Phosphate binds to the extracellular domain of CaSRs and, thus, keeps the CaSRs inactive [55]. Furthermore, acidosis influences the effects of CaMs on CaSRs. A study conducted by our research group revealed that the correction of acidosis reduced the PTH concentrations in patients undergoing dialysis [56]. Campion et al. demonstrated that acidosis directly inhibited CaSR activity in a histidine-independent manner [57]. Therefore, the phosphate control, along with the correction of metabolic acidosis, is crucial for improving the potency of CaMs in treatments for patients with CKD and ESRD.

#### 3.1.3. The Actions of CaMs beyond the Parathyroid Gland

In addition to direct CaSR activation, CaMs have other effects, which are listed as follows:

##### Increasing CaSR Insertion within the Plasma Membrane

Agonist insertion signaling (ADIS) contributes to the characteristic of a high degree of cooperation and absence of functional desensitization that characterizes the obvious CaSR signaling. ADIS occurs through agonist-driven release from pre-plasma membrane compartments and insertion at the plasma membrane, thus increasing and maintaining signaling. This agonist-mediated activation of CaSRs localized in the plasma membrane increases the rate CaSR insertion into the plasma membrane without altering the constitutive endocytosis rate, thereby considerably increasing the maximum signaling response. Prolonged CaSR signaling requires a large pool of intracellular ADIS-mobilizable CaSRs; this pool is sustained by signaling-mediated increases in biosynthesis [58].

##### Increasing Dihydroxy Vitamin D3 (1,25(OH)2D3) within the Kidney and Parathyroid Glands

CaMs can increase the activity of 1α-hydroxylase enzymes in the kidney and parathyroid cells in CKD. As a result, CaMs may increase the production of 1,25(OH)2D3 in the renal and parathyroid glands if the appropriate native vitamin D was present. By increasing the intracellular concentrations of 1,25(OH)2D3, CaMs can efficiently inhibit intra-gland PTH synthesis and secretion. Furthermore, CaMs can improve CaSR production and engender PTG proliferation inhibition [58].

##### Inhibiting FGF23 of Osteocytes

CaMs inhibit FGF23 production in bone cells, which may also improve klotho expression in parathyroid cells. Furthermore, the decrease in FGF23 can offset decreases in fibroblast growth factor receptor (FGFR) expression. Therefore, as FGF23 is reduced, the increase in FGFR and klotho expression can effectively inhibit PTH synthesis and secretion through a FGF-signaling cascade.

##### Attenuating Intestinal Calcium Absorption through TRPV6

CaSRs can be expressed in the intestinal epithelium, and CaSR activation attenuates TRPV6-dependent intestinal calcium absorption. However, chronic CaSR activation reduces the expression of genes involved in Ca^2+^ absorption. CaSR activation in the basolateral membrane of the intestine directly attenuates local Ca^2+^ absorption through TRPV6 to prevent hypercalcemia, which may explain how CaMs induce hypocalcemia [59]. 

**Figure 1 pharmaceuticals-15-00952-f001:**
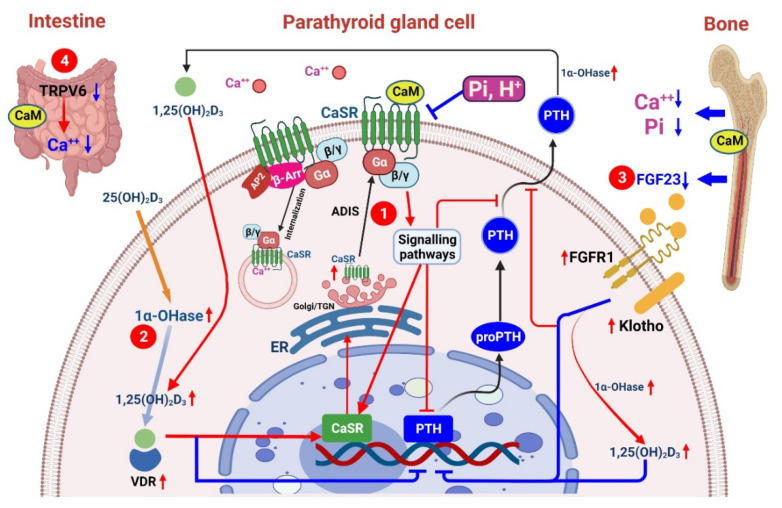
Effects of calcimimetics (CaMs) on parathyroid hormone (PTH) synthesis and secretion in chronic kidney disease (CKD). CaM compounds prevent increases in serum PTH concentrations and parathyroid hyperplasia in CKD and severe secondary hyperparathyroidism (SHPT). These changes occur despite decreases in serum 1,25-dihydroxy vitamin D3 (1,25(OH)2D3) and increases in serum phosphate, suggesting that calcium receptors play a prominent role in regulating parathyroid cell proliferation [60].

### 3.2. Effects of CaSRs on Renal Osteodystrophy

As CKD progresses, CaSR expression decreases within different tissues [40,61,62]. In advanced CKD, SHPT can activate PTH signal transduction on osteoblasts and eventually activate osteoclast-related bone resorption. This excessive bone resorption decreases the bone mass through the hyperactivity of osteoclasts. If PTH secretion is reduced, bone resorption can be alleviated. According to human studies, CaMs increased the serum bone formation markers, such as bone-specific alkaline phosphatase, in patients with SHPT or tertiary hyperparathyroidism but reduced the serum bone resorption markers [63,64,65]. In an in vivo study, CaMs maintained bone turnover while reducing PTH secretion [66]. CaMs reduce the release of calcium and phosphorous from the bones by inhibiting osteoclast bone resorption and stimulating osteoblast bone formation. In addition to reducing PTH secretion, CaMs might directly influence bone remodeling in osteoblasts and osteoclasts.

#### 3.2.1. Effect of CaMs on Osteoclasts

Osteoclasts are derived from monocyte/macrophages, and the differentiation process depends on RANK/RANKL signaling [67]. Although several reports have demonstrated that PTH might directly influence osteoclastogenesis, PTH is mainly considered [68] to influence osteoclastogenesis indirectly through RANK/RANKL activity in osteoblasts [69]. Excessive PTH concentrations increase osteoclast differentiation and, thus, aggravate bone resorption [70]. However, a study reported the preservation of osteoclast surfaces after CaM administration [66]. Moreover, the Homer1 complex interacts with CaSRs to enhance the mTORC2 and Akt pathways to maintain osteoblast proliferation and, further, osteocyte proliferation [71]. Zheng et al. demonstrated CaMs to exert a therapeutic effect on osteoclasts through Wnt-10b release, which will promote osteoblastogenesis [72]. CaSR inhibition influences the differentiation of osteoblasts by reducing the expression of essential proteins such as osteocalcin and alkaline phosphatase [73]. Furthermore, excessive PTH secretion may increase endoplasmic reticular stress within osteoclasts and thus activate autophagy [74]. CaMs may induce apoptosis and reduce excessive osteoclastogenesis, resulting from excessive oxidative stress due to SHPT [74]. When a high bone turnover occurs, CaMs induce bone formation and, thus, maintain the bone mass (Figure 2).

#### 3.2.2. Effects of CaMs on Osteoblasts

Wnt pathways are involved in the control of gene expression, cell behavior, cell adhesion, and cell polarity [75]. In these pathways, Wnt signaling inhibits β-catenin degradation, which can regulate the transcription of numerous genes [76]. Wnt signaling is activated through the ligation of Wnt proteins to their respective dimeric cell surface receptors that are composed of seven transmembrane Frizzled proteins and LRP5/6. Upon ligation to their receptors, the cytoplasmic protein disheveled (Dvl) is recruited, phosphorylated, and activated. Dvl activation induces the dissociation of GSK-3β from Axin and leads to GSK-3β inhibition [77]. β-catenin phosphorylation and degradation are then inhibited because of the inactivation of the “destruction complex”. Subsequently, stabilized β-catenin is translocated into the nucleus, leading to changes in the expression levels of various target genes [76].

Rodriguez et al. demonstrated that CaMs could increase the viability of osteoblasts. In nephrectomized mice with continuous PTH infusion, CaMs increased both the bone volume and osteoblast surface area. Moreover, increased nuclear Erk1/2 phosphorylation was observed in the osteoblasts, in addition to the increased expression of bone formation markers such as Runt-related transcription factor 2 (Runx2) and osteocalcin [66]. This finding demonstrates that CaMs influence bone formation through osteoblasts in SHPT (Figure 3). 

#### 3.2.3. Effects of CaMs on Bone Quality

Apart from the bone volume loss in SHPT, an unchanged bone volume may influence the bone quality. The bone structure is maintained by the deposition of calcium, inorganic phosphate, and type 1 collagen [78]. The crosslink between inorganic minerals and collagen contributes to bone strength. In CKD, both excessive oxidative stress and vitamin D deficiency influence bone strength by deteriorating collagen deposition. Advanced glycosylated end products may engender excessive oxidative stress in patients with CKD [79], and the deposition of nonenzymatic pentosidine in bone may influence the rate of bone formation and mineral apposition in patients undergoing dialysis [80]. In addition to inhibiting osteoclastogenesis, CaMs may affect bone strength. In adenine-induced CKD, the hardness and elastic modulus of bone were restored after cinacalcet supplementation [74]. Although the direct link between CaSRs and the expression of proteins associated with bone strength, such as type 1 collagen, is unclear, CaMs reduce oxidative stress in osteoclasts; therefore, they might be helpful in bone strength restoration.

#### 3.2.4. Role in CaMs in Conjunction with Vitamin D for Bone Remodeling

In low-bone-turnover disease, vitamin D deficiency directly reduces CaSR expression by downregulating these receptors. In the early stages of CKD, the therapeutic effects of CaMs may be less prominent than those of calcitriol in maintaining bone mass [81]. Finch et al. demonstrated that, in uremic rats used to investigate CKD Stage 3 or 4 in humans, the osteoid surface areas decreased with bone formation. This may be explained by the relatively low bone turnover in the early stages of CKD due to uremic toxin accumulation [20]. In other uremic animal models, CaMs played a therapeutic role in reversing high-bone turnover, along with, or independent of, PTH secretion control [66,82]. These in vivo studies have demonstrated that cinacalcet increased distal metaphysis more than calcitriol did in uremic rats [82]. Cinacalcet can induce the release of Wnt-10b from osteoclasts, which is associated with bone formation within osteoblasts [72]. Furthermore, cinacalcet could improve vitamin D activity during osteoclast differentiation. Allard et al. demonstrated that a therapeutic dosage of 1,25(OH)2D directly inhibited osteoclastogenesis without modulating bone resorption activity [83]. In CKD, the inhibitory effects of vitamin D were reported to decrease with the severity of CKD, and the resistance of osteoclasts to vitamin D could be reversed by cinacalcet [84]. According to these findings, CaMs, along with vitamin D, may reverse high-bone turnover by maintaining osteoblasts and inhibiting osteoclastogenesis in advanced CKD.

### 3.3. Effects of CaSRs on Vascular Calcification

Vascular calcification is common in advanced CKD because of renal osteodystrophy. High-bone turnover disease induces excessive bone resorption and thus increases calcium and phosphate deposition in extraosseous tissues. 

CaSRs are distributed within the endothelium, smooth muscle cells, and mesenchymal stem cells. Extracellular calcium can directly induce smooth muscle cell proliferation through mitogen-activated protein kinase (MAPK) kinase 1 [85]. A study reported that, in the uremic milieu, CaSR expression in the aorta decreased compared with that in a healthy control group [86]. Excessive calcium concentrations can downregulate the expression of CaSRs and the mineralization of vascular smooth muscle cells [87]. Decreased CaSR expression was reported to be associated with aortic calcification, with CaSRs serving as a proinflammatory cytokine. Mary et al. demonstrated that the expression of CaSRs in monocytes decreased in the uremic milieu, and the reduction of CaSR expression within the monocytes intensified vascular calcification [62]. In an animal model of spontaneous hypertension, CaSR expression was negatively associated with the plasma concentration of angiotensin II but positively associated with renin levels [88]. CaSRs may be distributed on vascular smooth muscle cells [85].

According to in vivo studies, CaMs alleviated vascular calcification and reduced PTH secretion in uremic rats [82]. Henley et al. compared the effectiveness of CaMs and calcitriol in alleviating vascular calcification. Calcitriol increased calcium and phosphate burdens by increasing calcium and phosphate reabsorption. As mentioned, excessive calcium intensifies vascular smooth muscle cell calcification by downregulating CaSRs [87]. Furthermore, the matrix Gla protein governing calciprotein particles is released under the influence of CaMs on vascular smooth muscle cells [89,90]. Additionally, nitric oxide might be directly released under CaM exposure, which could induce vasodilation and thus reduce the arterial pressure [91]. From the aspects above, the modulation of CaMs might play a role in alleviating vascular calcification. 

### 3.4. CaSRs and Left Ventricular Remodeling

CaSRs may be distributed within cardiomyocytes. CaSRs within cardiomyocytes induce the release of calcium from the intracellular endo/sarcoplasmic reticulum into the cytoplasm and thus engender cardiomyocyte apoptosis through mitochondrial damage [92]. During congestive heart failure, CaSR expression increases, along with mitochondrial membrane potential and ultrastructural change. Increased calcium/calmodulin-dependent protein kinase II and calcineurin expression heightens cardiomyocyte apoptosis and thus engenders further cardiac remodeling [93]. CaSRs are essential in modulating the renin–angiotensin–aldosterone system, and they govern juxtaglomerular renin release [94]. CaMs alleviate aortic calcification directly in the vascular smooth muscle cells by inhibiting MEK1 [85]. In contrast to extracellular calcium, CaMs may not influence CaSR expression [95]. CaSR activation could induce myocardial apoptosis by modulating the mitochondrial dynamics. Hong et al. demonstrated that the myocardial expression of CaSRs increased in an animal model of spontaneous hypertension [96]. In the animal model, mitochondrial fission increased in the cardiomyocytes, and mitochondrial swelling, disorganized cristae, and a loss of normal striation activated downstream apoptosis. Generally, CaSR inhibitor administration can alleviate cardiac hypertrophy through multiple mechanisms. CaMs may play a protective role in alleviating changes in the mitochondrial membranous potential and the release of calcium/calmodulin II [93]. The direct modulation of CaSRs on the renin–angiotensin–aldosterone system (RAAS) also alters cardiac remodeling [97]. Additionally, CaM administration can reduce unwanted remodeling by ameliorating autophagy [98]. Therefore, CaMs should be a therapeutic choice under conditions involving vascular calcification. 

### 3.5. Role of CaSRs in Renal Tubules and Vasculature

CaSRs are distributed within different tissues, including those of the heart, pancreas, and renal tubules. In proximal tubular cells, CaSRs are located on the luminal side. Filtered calcium on the luminal side directly activates CaSRs, thus leading to sodium/proton exchanger isoform 3 activation within the luminal side. Consequently, protons generated within the renal tubules can be extruded into the lumen and engender a net reabsorption of bicarbonate [99]. In the thick ascending tubules, CaSRs are located on the basolateral side. CaSR activation modulates the intracellular calcineurin–nuclear factor of activated T-cell signaling, which reduces paracellular calcium reabsorption by downregulating claudin expression [100]. Rothe et al. reported that the CaSR polymorphism Arg990Gly was associated with a more favorable response to CaSRs in transfected HEK cells [101]. In addition, CaSRs within the juxtaglomerular apparatus can modulate the release of renin. The interaction between CaSRs and RAAS may directly modulate the Na^+^/Ca^2+^ exchanger [102]. Ortiz-Capisano et al. revealed that CaM activation increased the intracellular calcium release through phospholipase C/inositol triphosphate3 and RyR activation; hence, the release of vasoactive renin could be inhibited [94]. The chronic activation of CaSRs could alter RAAS and lead to hyperreninemia [103]. The interaction between CaSRs and RAAS may be essential for treating cardiac remodeling in patients with CKD–mineral and bone disorder.

## 4. Clinical Efficacy of CaMs

### 4.1. CaMs in Renal Osteodystrophy/Fracture

In patients with primary hyperparathyroidism, CaMs provide a therapeutic option. Peacock et al. demonstrated that cinacalcet administration could lower the PTH concentrations in patients with primary hyperparathyroidism. In addition, bone resorption and formation markers both increased after treatment, without any changes occurring in the bone mineral density. Therefore, cinacalcet can successfully reduce PTH concentrations and improve the serum calcium and phosphate control [104], and hypercalcemia can be corrected under such treatment [105]. Moreover, the OPTIMA study, which compared the effects of cinacalcet with those of vitamin D, demonstrated that cinacalcet provided a more favorable PTH control than vitamin D [106]. Therefore, the combination of vitamin D with CaMs can be a rational strategy for treating or controlling SHPT. The ACHIEVE study revealed that CaMs, along with low-dose vitamin D, provided better control of SHPT and decreased PTH levels compared with the other treatment approaches [107]. The DUET trial demonstrated that CaMs, along with vitamin D, exhibited superior PTH control than vitamin D alone or CaMs with calcium supplements [108].

Scholars have demonstrated the clinical efficacy of CaMs concerning bone mineral density in patients with SHPT. For example, Shigematsu et al. reported that, during a 1-year cinacalcet treatment period for patients with hemodialysis, the bone resorption markers decreased with the transient elevation of bone-specific alkaline phosphatase [109]. Tsuruta et al. performed a clinical trial and reported that, in patients with HD with SHPT, a 1-year administration of cinacalcet engendered bone mineral density improvement, which was negatively correlated with variations in the PTH concentrations [110]. Specifically, the observed bone mineral density increased and serum alkaline phosphatase concentrations deteriorated. Hung et al. indicated that cinacalcet (25 mg/day) supplementation for 6 months increased bone mineral density in patients with HD. Among the patients who demonstrated a response to treatment, bone formation markers such as procollagen type I N-terminal pro-peptide (PINP) increased, but bone resorption markers such as Tartrate-resistant acid phosphatase isoform 5 regressed. Furthermore, plasma Wnt-10b increased in the cinacalcet-responsive group [65]. Ruderman et al. demonstrated the negative effects of cinacalcet withdrawal on the plasma parameters. Based on the aforementioned results, in SHPT with clear bone resorption, CaMs play a therapeutic role in maintaining the bone mass.

### 4.2. CaMs in Cardiovascular Mortality

The role of CaMs in cardiovascular disease was first elucidated in the EVLOVE study. In this study, the 21.2-month administration of cinacalcet did not lower the overall cardiovascular mortality in patients with HD with moderate-to-severe SHPT [111]. In a secondary investigation, the number of cardiovascular events mediated by nonatherosclerotic disease and the number of events of sudden death or congestive heart failure decreased in patients who received cinacalcet [48]. Furthermore, CaSR-associated polymorphisms influence the effectiveness of PTH reduction. For example, the rs9470 polymorphism was associated with more favorable reductions of PTH concentrations in patients who received cinacalcet treatment [112]. Another study revealed that a CaSR polymorphism was associated with dyslipidemia in patients with HD; however, the clinical relevance of this finding is unknown [113]. Accordingly, CaMs may play a therapeutic role in alleviating cardiovascular disease associated with non-dyslipidemia-related risk factors.

Ruderman et al. observed that primary calciprotein particles in plasma, which are negatively associated with vascular calcification and PTH concentrations increased 1 year after the cessation of cinacalcet treatment in patients with ESRD [114]. Although the clinical relevance of this change in calciprotein particles after cinacalcet withdrawal remains unclear, the production of primary calciprotein particles may alleviate the calcification burden after the cessation of cinacalcet treatment.

### 4.3. CaMs Compared with Parathyroidectomy

In SHPT, the diffusion of hyperplasia in the parathyroid glands transforms polyclonal hyperplasia into monoclonal hyperplasia. Thus, single nodular hyperplasia is common among patients with advanced CKD or those receiving dialysis [115]. Subtotal or total parathyroidectomy is a curative strategy for treating uncontrolled SHPT or SHPT with severe calciphylaxis. Only a few clinical trials have compared the efficacy of CaMs and parathyroidectomy. Keutgen et al. conducted a comparative study, in which the endpoint was bone mineral density improvement. The surgical group demonstrated greater PTH improvement than the CaM group; this PTH improvement was associated with a greater bone mineral density improvement regardless of the choice of treatment [116]. From the study by Cruzado et al., parathyroidectomy provided a better control of hypercalcemia than cinacalcet [117]. Almadén et al. reported a decreased expression of CaSRs in hyperplastic nodules and thus recommended surgical management in tertiary hyperplasia [118], which may result in lower responses to cinacalcet treatment. Nevertheless, surgical intervention may induce a relatively high incidence of hypocalcemia. To achieve sustained PTH secretion control and pleiotropic effects through CaSRs in SHPT, the application of CaMs is an essential therapeutic strategy.

### 4.4. Current Guidelines on Treating CKD–MBD with Calcimimetics

Based on the KDIGO guidelines on the treatment of CKD–MBD, secondary hyperparathyroidism in ESRD subjects could be treated by the CaMs, calcitriol, or vitamin D analogs or a combination of CaMs with calcitriol or vitamin D analogs [1]. Both vitamin D analogs and CaMs are acceptable options for treating SHPT. For patients with both hypercalcemia and hyperphosphatemia, CaMs is a reasonable choice in lowering the PTH level in comparison with the vitamin D analog. As hypocalcemia develops, the CaMs should be discontinued, and the vitamin D or vitamin D analog could be the agent lowering the PTH [119]. Recent clinical trials about the new generation of CaMs have provided the PTH-lowering effect. However, the effect of fracture prevention and the cardiovascular outcome in other CaMs should be noticed in the future.

## 5. Limitation of the CaMs in CKD–MBD

Based on the evidence from clinical trials such as the EVOLVE study, CaMs provided a sustained PTH-lowering effect and therefore might contribute to the improvement in cardiovascular mortalities in previous sections. However, there are still several limitations to use in CaMs. First, the serum calcium concentration should be monitored continuously to avoid hypocalcemia. Second, the cost of CaMs might pose an economic burden for the patients. Third, the trials for a head-to-head comparison between CaMs and parathyroidectomy are still lacking. Further evidence is needed to compare the efficacy between CaMs and parathyroidectomy.

## 6. Conclusions

CKD–MBD is a severe complication in patients with CKD and ESRD. Decreased renal phosphate excretion and SHPT increase bone resorption and not bone formation, thus intensifying the vascular calcification and left ventricular remodeling. Decreases in CaSR expression within the parathyroid glands and vasculature potentiate the hazards of SHPT because of the corresponding vitamin D deficiency and dysregulation of the proteins for cell cycling. To alleviate these effects, CaMs can be combined with vitamin D to treat CKD–MBD.

## Figures and Tables

**Figure 2 pharmaceuticals-15-00952-f002:**
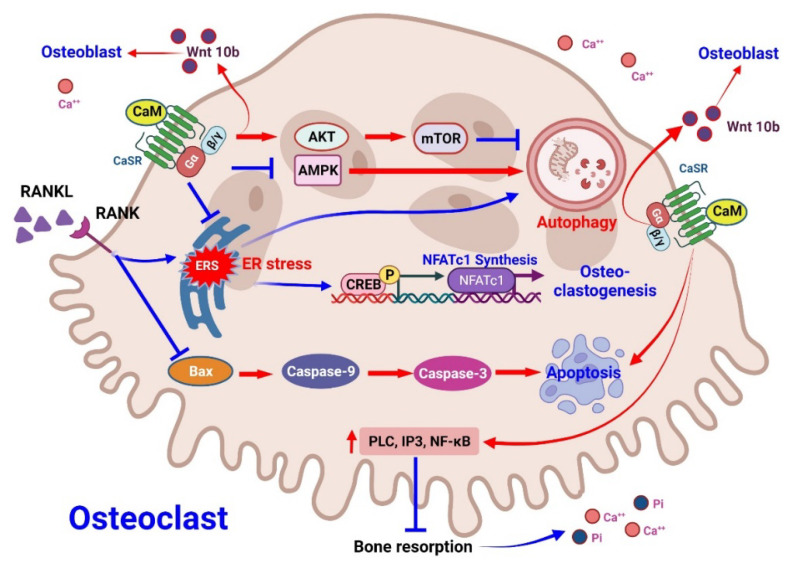
Molecular mechanism of CaMs on osteoclasts in CKD. Osteoblast and osteoclast proliferation, differentiation, and apoptosis are influenced by local extracellular calcium concentrations. The activation of the RANK/RANKL signal from osteoblasts modulates osteoclastogenesis. The signal induces the elevation of endoplasmic reticulum stress (ER stress), which activates the cAMP response element-binding protein (CREB) phosphorylation. The downstream transcription of the nuclear factor of activated T cells, cytoplasmic (NFATc1) activates osteoclastogenesis. On the other hand, ER stress also activates the autophagy process to maintain osteoclast viability. RANK/RANKL also inhibits osteoclast apoptosis. CaMs induce the release of Wnt-10b from osteoclasts and inhibit the autophagy vial, activating the Akt/mTOR pathway. In this manner, CaMs increase bone formation, along with the apoptosis of the osteoclast.

**Figure 3 pharmaceuticals-15-00952-f003:**
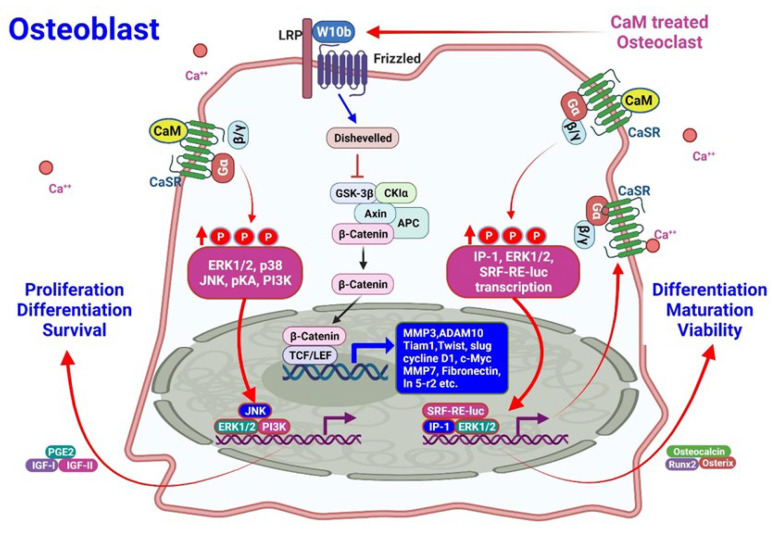
The molecular mechanism underlying the effects of CaMs on osteoblasts in CKD. The Wnt-10b released from osteoclasts by CaMs activates β-catenin by inhibiting GSK-3β-mediated degradation. On the other hand, nuclear Erk1/2 phosphorylation activated by CaMs activates the transcription, along with c-Jun N-terminal kinase (JNK)/PI3K, which increases osteoblast proliferation and maintains survival. The activation of Erk1/2 with serum response factor-response element (SRF-RE)-luc and insulin-like growth hormone factor is responsible for osteoblast proliferation/survival and differentiation and its maturation.

## Data Availability

Not applicable.

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
