# Peer review of "Role of Calcimimetics in Treating Bone and Mineral Disorders Related to Chronic Kidney Disease"

_pharmaceuticals, 2022, doi:10.3390/ph15080952_

Round 1

Reviewer 1 Report

The manuscript pharmaceuticals-1820412 by Yi-Chou Hou et al., entitled “Role of calcimimetics in treating Chronic kidney disease-mineral and
bone disorders” is an interesting molecular and clinical review focused on the role of calcimimetics in renal osteodystrophy.

 The manuscript is written with a flow, smooth and understandable language.

The introduction is complete and provides sufficient background.

The figures are designed appropriately.

The figure legends describe very understandable the reported figures and are explicit in the description.

Just a minor about the format set of the text, for instance, in lane 97, and in other part of the manuscript, both title of the paragraph and the text are in italic. Furthermore, the figure legends need to be distinguished from the manuscript text.

Lane 96: “rates..”, please correct the punctuation

Lane 100: “proteins), increases . because of the release”, please correct the punctuation.

Lane 134: ”osteeodystrophy “, please correct

Lane 187: please, insert the full stop

In my opinion, the manuscript could be accepted for publication in the Pharmaceuticals journal after the minor adjustments proposed above.

Reviewer 2 Report

This is a narrative review which focuses on the role of CaMs in renal osteodystrophy, including their mechanisms and clinical efficacy

It’s a very comprehensive review in this field

However I suggest to shorten it in some points ( from 2.3 forward)  and to perform a linguistic revision

Reviewer 3 Report

The paper is very well written and provides comprehensive information on the pathophysiology of bone and mineral disorders related to chronic kidney disease as well as mechanisms of action and effects of calcimimetics. 

However, there are some issues which should be corrected prior publication:

1. The title of the paper is a little bit unclear; I would suggest change into: "Role of calcimimetics in treating bone and mineral disorders related to Chronic kidney disease"

2. Figures 1-3 are positioned at the beginning of sections 2.1, 2.2.1 and 2.2.2, respectively. Figures should be positioned after the text in these paragraphs and not at the beginning of paragraphs

3. Title of the section 2.1. is Action of CaMs (Figure 1) - remove reference to the figure from the section title.

4.  "." and ":" are not used in the titles of the sections. Please remove them from titles of section 2.1.1., 2.1.2, 2.1.3., 2.1.4, 2.2.1. and 2.2.2

5. Section 1.1., 1.3 starts with some "introduction" in italic while section 2.1.1. is completely written in italic. Can you clarify why these parts of the text are written in italic? I would suggest to use normal font.

Round 2

Reviewer 3 Report

The Authors have corrected manuscript according to the recommendations of the reviewers and improved the manuscript

This manuscript is a resubmission of an earlier submission. The following is a list of the peer review reports and author responses from that submission.